# Functionalization Mechanism of Reduced Graphene Oxide Flakes with BF_3_·THF and Its Influence on Interaction with Li^+^ Ions in Lithium-Ion Batteries

**DOI:** 10.3390/ma14030679

**Published:** 2021-02-02

**Authors:** Łukasz Kaczmarek, Magdalena Balik, Tomasz Warga, Ilona Acznik, Katarzyna Lota, Sebastian Miszczak, Anna Sobczyk-Guzenda, Karol Kyzioł, Piotr Zawadzki, Agnieszka Wosiak

**Affiliations:** 1Institute of Materials Science and Engineering, Lodz University of Technology, Stefanowskiego 1/15, 90-924 Lodz, Poland; Lukasz.Kaczmarek@p.lodz.pl (Ł.K.); magdalena.balik@edu.p.lodz.pl (M.B.); sebastian.miszczak@p.lodz.pl (S.M.); anna.sobczyk-guzenda@p.lodz.pl (A.S.-G.); piotr.zawadzki@p.lodz.pl (P.Z.); 2Łukasiewicz Research Network—Institute of Non-Ferrous Metals Poznań Division, Forteczna 12, 61-362 Poznan, Poland; ilona.acznik@claio.poznan.pl (I.A.); katarzyna.lota@claio.poznan.pl (K.L.); 3Department of Physical Chemistry and Modelling, Faculty of Materials Science and Ceramics, AGH University of Science and Technology, Mickiewicza 30, 30-059 Krakow, Poland; kyziol@agh.edu.pl; 4Institute of Information Technology, Lodz University of Technology, Wólczańska 215, 90-924 Łódź, Poland; agnieszka.wosiak@p.lodz.pl

**Keywords:** graphene, graphene oxide, graphene doping, lithium-ion batteries

## Abstract

Doping of graphene and a controlled induction of disturbances in the graphene lattice allows the production of numerous active sites for lithium ions on the surface and edges of graphene nanolayers and improvement of the functionality of the material in lithium-ion batteries (LIBs). This work presents the process of introducing boron and fluorine atoms into the structure of the reduced graphene during hydrothermal reaction with boron fluoride tetrahydrofuran (BF_3_·THF). The described process is a simple, one-step synthesis with little to no side products. The synthesized materials showed an irregular, porous structure, with an average pore size of 3.44–3.61 nm (total pore volume (BJH)) and a multi-layer structure and a developed specific surface area at the level of 586–660 m^2^/g (analysis of specific surface Area (BET)). On the external surfaces, the occurrence of irregular particles with a size of 0.5 to 10 µm was observed, most probably the effect of doping the graphene structure and the formation of sp^3^ hybridization defects. The obtained materials show the ability to store electric charge due to the development of the specific surface area. Based on cyclic voltammetry, the tested material showed a capacity of 450–550 mAh/g (charged up to 2.5 V).

## 1. Introduction

Lithium-ion batteries are the subject of intensive research due to their high potential for applications [1]. Li-ion cells are the main source of power for portable electronic devices such as mobile phones [2], notebooks and tablets [3]. The possibility of their use in electric vehicles has also been appreciated [4].

In most cases, Li-ion batteries are made of anode material in the form of graphite (C) and liquid lithium compounds as an electrolyte (e.g., C/LiPF_6_ [5]). Unfortunately, graphite anodes, due to their low theoretical specific capacity (372 mAh·g^−1^) [6], high molar volume [7], limited volumetric energy density stored in the battery and extended charging time, significantly limit their potential for applications requiring higher power and energy densities for the power source [8]. An important role is played by such parameters of the electrodes as electronic/ion conductivity, molar mass, redox potential [5].

Due to the growing requirements for Li-ion cells, extensive research is carried out on the development of new electrode materials that will ensure higher energy density, high efficiency and a longer cycle life [9]. One of them is carbon-based materials, such as carbon nanotubes [10,11] or graphene [12]. The latter is especially interesting for the production of anode for lithium-ion batteries (LIB), due to its unique, two-dimensional (atomic) structure [13], high theoretical specific surface (2630 m^2^*g^−1^) [14] and good electrical conductivity [15], which enables the free flow of electrons and electrolyte ions, and thus the intercalation and deintercalation of lithium ions [16].

Unfortunately, in most cases the analyses are only related to computer simulations. In fact, the non-defected structures of graphene are virtually impossible to produce with the current advancement of synthesis methods and techniques. The vast majority of research on functionalization concerns graphene oxide (GO), and not graphene itself. The presence of oxygen groups in the graphene structure significantly lowers its electrochemical properties [17].

Meanwhile, the controlled induction of disturbances in the graphene lattice allows for the production of numerous active sites for lithium ions on the surfaces and edges of graphene nanolayers and the improvement of graphene functionality in LIBs [18]. Doping is one of the most effective and widely used defect induction methods [19]. According to literature reports, fluorine can be successfully used as a dopant in the process of graphene functionalization, because it is the most electronegative element, capable of binding with carbon and creating exceptionally strong single covalent bonds (488 kJ * mol^−1^) [20].

In recent years, the number of studies and publications on the fluorination of graphene oxide (GO) and reduced graphene oxide (rGO) has increased. One of them is the gas fluorination process with the use of such precursors as F_2_ [21], or other compounds based on fluorine: SF_6_, SF_4_, MoF_6_ [22]. Moreover, GO can be subjected to plasma [23], photochemical [24] and electrochemical [25] fluorination. Very satisfactory results are achieved during GO fluorination in hydrothermal or solvothermal processes with the use of hydrofluoric acid (HF) [26], boron trifluoride BF_3_ [27] and other fluorine-containing reagents [4,28].

In his publication, Haoran described the hydrothermal process of producing a hydrogel based on graphene, doped with fluorine atoms. GO was used as the starting material, the source of fluorine was hydrofluoric acid. The mixture was dispersed by ultrasound and then subjected to an elevated temperature (90 °C, 120 °C, 150 °C and 180 °C) for 24 h. The obtained material was then immersed in deionized water to remove residual acid and then dried at room temperature. Studies have shown that the fluorine content can be easily controlled by the temperature of the process. The XPS and FTIR results mainly indicated the presence of semi-ionic CF bonds that facilitate ion transport, improve electrical conductivity and provide active sites for lithium ions. The highest electric capacitance (227 F * g^−1^) and the highest efficiency were recorded for the material obtained in the hydrothermal process carried out at the temperature of 150 °C [29].

Damien and his team conducted and described the process of making the F-GO. The starting material used was a fluorinated graphite polymer ((CF_0.25_)_n_) which was dispersed in a mixture of H_2_SO_4_–H_3_PO_4_ (ratio 9:1) acids. Then it was stirred for 2 h at 50 °C. An amount of 18 g of KMnO_4_ was added to the mixture, and then 10–12 mL of H_2_O_2_. The obtained F-GO was subjected to the reduction process by adding four drops of hydrazine monohydrate and heating under a vacuum at 90 °C for 3 h. The material obtained this way showed a specific capacity of 767 mAh * g^−1^ at a current density of 10 mA * g^−1^ [4].

The results presented in the literature show that the produced F-GO and F-rGO nanoparticles can be successfully used for the production of high-performance lithium-ion batteries. However, no attempts were made to systematize the research, which effectively limits the possibility of optimizing the capacity of lithium-ion batteries based on graphene structures.

This work summarizes the numerical analyses (using molecular modeling in the SCIGRESS v.FJ 2.7 program) and experimental studies of the physical and electrochemical properties of doped (modified) graphene structures in order to determine the type and properties of groups formed as a result of the boron fluoride tetrahydrofuran (BF_3_·THF) reaction with rGO flake. This will allow understanding of the mechanisms taking place during the doping of graphene with fluorine and to determine the influence of the performed functionalization on its interaction with Li^+^ ions.

## 2. Materials and Methods 

rGO doped with boron and fluorine atoms was obtained in a hydrothermal process. An amount of 1 mg rGO (Sigma-Aldrich, Stainheim, Germany) was dispersed in 100 ml of distilled water using an ultrasonic sonicator (Sonics VCX130, Sonics, Newtown, MA, USA). Then, solutions were prepared from the suspension thus obtained by adding 1.5 ml and 3 ml of BF_3_ solution (1.0 M BF_3_ in THF, Sigma-Aldrich, Stainheim, Germany), respectively, and heating it in a water bath at 100 °C for 24, 48, 72, 96 or 120 h (Table 1). The resulting suspension was filtered off and dried in a nitrogen atmosphere at 20 °C for 24 h.

The simulation of phonon vibration spectra of the functionalized graphene was performed as follows. In order to determine the actual functional groups present in the structure of a graphene flake, the appropriate infrared spectra were simulated using the SCIGRESS v.FJ 2.7 software. For this purpose, as a model system of atoms, a graphene structure was created, consisting of 149 carbon atoms and 46 hydrogen atoms, which saturate carbon bonds at the edge of the analyzed graphene flake. Additionally, a defect in the form of a vacancy of one C atom was generated in this structure. This structure represents the actual molecular structure of the flake graphene after the reduction process. Then the created system was optimized in order to achieve the energy minimum. For the obtained energy-optimized structure, molecular modeling was carried out using the semi-empirical method of quantum computing (MOPAC: MO-G PM3), which allowed the identification of the interaction of infrared radiation with characteristic chemical groups.

Infrared absorption spectra of the graphene samples in the spectral range 4000 to 400 cm^−1^ were collected using an Nicolet iS50 Fourier-transform IR spectrometer (Thermo Fisher Scientific, Waltham, MA, USA). Spectra were recorded with the resolution of 2 cm^−1^ using a high sensitivity MCT-B detector (mercury cadmium telluride). The measurements were performed in a reflection mode with an application of a Sequelle DRIFT accessory working at an angle of incidence equal to 20 degrees. In each case, data from 128 scans were collected to construct a single spectrum.

Raman studies were conducted using a Horriba Labram HR (HORIBA Jobin Yvon) micro-Raman spectrometer. Laser power of the 532-nm excitation source (Nd:YAG green laser) was set to 0.3 mW (1%) and 3 mW (10%).

The morphology and microstructure of graphene material was characterized using a scanning electron microscope (SEM) Jeol JSM-6610 (JEOL Ltd., Tokyo, Japan) with an EDS system. The observations were carried out in secondary electron imaging mode (SEI), with an accelerating voltage of 20 kV.

Textural properties of the samples were determined using Micromeritics ASAP 2020 equipment. The total specific surface area (TSSA) analysis was based on the analysis of specific surface area (BET) model of N_2_ low temperature adsorption and assumption that nitrogen molecules cover 0.162 nm^2^ of adsorbent surface. Size and volume of pores between 1.03 nm and 67.5 nm radius were determined using total pore volume (BJH) desorption cumulative volume of pores and BJH desorption average pore radius. During the analysis, ca. 0.1–0.3 g of the samples was placed in a measurement ampoule and degassed for 4 h at 100 °C. Then, the ampoule was attached to the instrument and the adsorption process was carried out at −195 °C.

## 3. Results and Discussion

### 3.1. Molecular Analysis

According to the simulation and optimization of the analyzed graphene structure carried out in the SCIGRESS v.FJ 2.7 program, the C-C bond length is 1.42 Å, while the C-C-C torsion angle in the undefected area is 120°. For such an energetically optimized structure, the system energy simulations were carried out depending on the place of attachment of the -BF_2_ group or the formation of -BH_2_ groups. Designations of the analyzed graphene systems depending on the place of attachment of -BF_2_ groups or formation of -BH_2_ groups are summarized in Table 2.

First, the analysis covered the interaction between the -BF_3_^-^ ion and a model graphene flake composed of 149 carbon atoms and 46 hydrogen atoms with a structural defect in the form of a vacancy of one C atom. The interaction energy of the -BF_3_^-^ ion as a function of the distance from the surface of the unmodified, defective graphene flake (Gdef.) decreases from 406 to 230 kcal/mol.

Then, the energy values of the most probable systems for which the addition of BF_2_ group or the formation of -BH_2_ groups may take place were analyzed. Based on the analysis of the energy values of graphene systems depending on the mentioned parameters, it was determined that the most thermodynamic stable structures are the ones in which, in the reaction of BF_3_·THF with graphene, -BF_2_ groups are formed on its surface (Table 3, Figure 1, Figure 2). On the other hand, energetically, there are no privileged reactions leading to the formation of in-atomic systems in relation to which the carbon atoms of the graphene structure are substituted with boron (the energy of the system is 413 kcal/mol). The energy of the analyzed defective graphene (vacancy of 1 C atom), which is the reference for the calculations, is 465 kcal/mol. Attachment of the -BF_2_ groups both to the edge of the flake (system energy equals 283 kcal/mol) and to carbon with hybridization of sp^3^ or sp^2^ in the area of graphene defect (system energy equals 296 kcal/mol), and also adatomically (to carbon from graphene structure with sp^2^ hybridization, with system energy equalling 297 kcal/mol) is practically on the same level.

In the areas of graphene where the reaction with the BF_3_^-^ ion produces the -BF_2_ group, there is a growth of negative zones. Additionally, there is a clearly marked tendency to occupy the central zones of the graphene flake by a negative charge. A similar relationship was found by X. Duan, K. O’Donnell [30], who analyzed graphene systems doped with sulfur and/or nitrogen.

Then, for the most thermodynamically stable system (G def.-Bad-atom-F-described in Table 2), taking into account the electrostatic relations, the interaction energy with the lithium ion as a function of its distance from the graphene structure was examined (Figure 3). It was assumed that the lithium ion would only interact with the structure of functionalized graphene. In this case, the transport of the shielded lithium through the electrolyte and its interaction with the SEI layer of the lithium-ion battery was omitted.

For such assumptions, it was found that the change in the electrostatic potential of the functionalized graphene flake (C_graphene_-BF_2_ chemical bond) causes the physical interaction of graphene with the lithium cation. The consequence of this phenomenon is the reduction in the system energy as a result of the electrostatic attraction of Li^+^ by the areas of electrostatic interaction of fluorine. This phenomenon explains why it is possible to achieve a relatively thermodynamically stable physical lithium bond in modified graphene systems—as opposed to unmodified graphene systems. Although the energy of the system decreases as Li^+^ approaches the graphene surface, the minimum value of the energy of the system is close to 670 kcal/mol compared to G_def._-B_ad-atom_-F, whose energy is about 503 kcal/mol.

### 3.2. FTIR

In order to identify functional groups present in the research material after the functionalization process, FTIR analysis was performed.

Figure 4 shows the FTIR spectra of rGO without modification as well as after functionalization with BF_3_·THF. In the spectrum of pure, unmodified rGO there is a broad band with a maximum at the wavenumber of 3440 cm^−1^, which comes from the stretching vibrations of the -OH groups. This bandwidth for the modified rGO samples is slightly lower. The deformation vibrations of the -OH groups are also visible at the wave number of 1450 cm^−1^. Another clear peak in the unmodified rGO spectrum is the peak at 1720 cm^−1^, which comes from the stretching vibrations of the C=O bond, which may indicate that, despite the GO reduction process, some carbon–oxygen bonds still remained in its structure [31].

The peak characteristic for the graphene structure is located at 1580 cm^−1^, which comes from the stretching vibrations of C=C bonds [32]. Increasing the intensity of this absorption maximum for rGO after BF_3_·THF functionalization may result from the overlapping of π–π bonds in the so-called stacks. Then in the spectrum of unmodified rGO there is a wide, flat band in the range of 1400–950 cm^−1^. In this case, the components are mainly -OH and C=C bonds, characteristic for a material of this type. In turn, for this range, the most significant differences in the course of the spectra were observed for samples after BF_3_·THF functionalization. Additional, distinct peaks appeared at 1310–1280, 1126, 1085, 1055, 1035 and 1018 cm^−1^.

The band in the 1310–1280 cm^−1^ range comes from the stretching vibrations of the bonds of the CF-CH_3_ terminal group [33] and the stretching vibrations of the asymmetric C-F bonds in the CF_2_ group [34]. The wide band with a maximum of 1120 cm^−1^ comes from symmetrical stretching vibrations of the C-F bonds also belonging to the CF_2_ group [34,35]. Another confirmation of the presence of C-F bonds can be found in the peaks located at 1085 and 1035 cm^−1^, which come from the fragment of the structure in which the carbon atom is connected to only one fluorine atom [36]. It should be remembered that with the wavenumber value of 1085 cm^−1^, the B-OH stretching vibrations may also have their maximum absorption [37]. In the spectra of some rGO samples there is a weak peak at 1055 cm^−1^, which comes from the ether bond fragment belonging to the not fully dissociated tetrahydrofuran [36].

Table 4 shows the areas of the above-discussed peaks. The obtained data show that the highest intensity of the peaks coming from vibrations of CF bonds adjacent to the methyl group have the samples modified with BF_3_·THF with a concentration of 1.5% for the 24- and 36-h bath times, and with a higher concentration of 3%, the stronger effect was obtained after 16 h. In this range of wavenumbers there may also be vibrations of B-F bonds constituting “contamination” after the used modifier [38]. In the case of the total amount of C-F bonds contained in CF_2_ and CF groups, for a modifier concentration of 1.5%, the times in the range of 16–32 h are promising. A further increase in the functionalization time does not cause a further increase in the number of groups containing fluorine atoms in their structure. A slightly different relationship was obtained for the samples functionalized with a concentration of 3%. In this case, extending the duration of this process led to a gradual increase in the amount of attached fluorine. The content of impurities in the form of C-O-C groups changes randomly. In this case, no dependence on the concentration of BF_3_·THF or the time of modification was observed. With the wavelength value of 1018 cm^−1^, another peak appeared, which confirms that the graphene structure was not only doped with fluorine atoms, but also an addition of boron atoms. The peak next to this wavenumber value clearly proves that there are also C-B bonds [39]. Its amount in each sample undergoes similar changes as the content of fluorine. Both spectra also distinguish a broad band with a maximum of 720 cm^−1^, which additionally confirms the presence of the -CF_2_ group. In this case, the deformation vibrations of the C-F bond are present.

### 3.3. Raman Spectroscopy

Further characterization of the structure of the obtained materials was performed using Raman spectroscopy. On the spectrum obtained for the starting material, which was rGO, peaks characteristic of graphene structures were observed: peaks D (1342 cm^−1^), G (1580 cm^−1^) and 2D (2698 cm^−1^) (Figure 5).

The D-band corresponds to the disorder of the structure due to the disturbance of the symmetry of the graphene lattice, the presence of sp^3^ hybridization-based defects, vacancies, grain boundaries and even edges. It comes from the secondary Raman scattering process involving one iTO phonon and one defect [40]. In the case of the parent graphite, it has a relatively low intensity, which proves the highly crystalline structure of the tested sample [41].

On the other hand, the shape and intensity of the G-band result from the vibration of carbon atoms with sp^2^ hybridization in a two-dimensional hexagonal lattice. The G applies to all carbon structures with sp^2^ hybridization, including amorphous carbon, carbon nanotubes, graphite, etc. [42]. It is the result of photon scattering on the optical phonon and comes from the main, primary Raman scattering [43].

The 2D-band comes from the secondary Raman scattering process involving two iTO phonons near the K point [40]. On the Raman spectra of the functionalized structures, the delamination of the G peak (D’) and the appearance of the D + G peak can also be observed.

The process of rGO functionalization with the use of BF_3_·THF significantly influenced the changes in individual bands. The G peak was observed at a wavelength of 1566 cm^−1^; there was a shift towards lower wavenumbers relative to the peak position for the unmodified sample. We observe an increase in its frequency with a reduction in the FWHM (full width at half maximum). It comes from the fact that the doping of the graphene structure causes the Fermi energy to move away from the Dirac point, and the plasma–phonon coupling effect weakens, which is manifested by an increase in the G phonon energy [40].

The number of defects in the graphene structure, increasing with the progress of the functionalization process, was manifested by an increase in the intensity of the D, D’ and D+G peaks, observed at 1339, 1596 and 2461 cm^−1^.

The D ’peak becomes visible on the Raman spectrum if the graphene material contains randomly distributed impurities or surface charges. This is due to the fact that the localized vibration modes of contaminants can interact with the extended phonon modes of graphene, which results in the observed division [44].

The presence of the D peak is related to the disorder and defect of the graphene layer. In the starting material, it is the result of the presence of oxygen functional groups. The functionalization process leads to an increase in the intensity of the D band, which is related to the reactions taking place and the formation of sp^3^ hybridization bonds.

The D and D ’bands are created as a result of the photon scattering on iTO phonons from the vicinity of the K point of the Brillouin zone and the iLO from the vicinity of the point Γ Brillouin zone. In order to fulfill the principle of conservation of momentum, the proportion of defects that take over this excess momentum without changing the energy is necessary. Thus, the analysis of the intensity of the D and D’ bands allows the determination of the level of damage to the carbon layer [45].

All types of sp^2^ hybridization carbon materials show a strong peak in the range 2500–2800 cm^−1^ in the Raman spectrum. In combination with the G-band, this spectrum is the Raman signature of graphite materials and is called 2D [46]. The 2D band is a second-order two-phonon process and shows a strong dependence of frequency on the energy of the excitation laser. It is also used to identify a single layer of graphene by examining the half-width and the ratio of the intensity of 2D and G bands. For a single layer of high-quality (defect-free) graphene, it should equal 2 and the half-width should be close to ~30 cm^−1^. In the case of a greater number of layers, the 2D band becomes wider, while the ratio of the intensity of the 2D band to G is lower than one [47].

The I_2D_/I_G_ ratio for the starting material is 0.62 and for modified samples it ranges from 0.56 to 0.69. The half-width for the rGO before the modification process is 203.6 cm^−1^, and for successively functionalized materials 127.93 cm^−1^ (A1), 135.84 cm^−1^ (A2), 114.73 cm^−1^ (A3), 123.89 cm^−1^ (A4), 146.59 cm^−1^ (A5), which confirms the fact that the tested material consists of more layers.

Based on the presented data in Table 5, we can observe that the 2D shifts towards lower wavenumbers, which indicates the presence of tensile stresses [44].

Figure 5 shows the Raman spectra for the rGO material before modification and the obtained end products (A1–A5).

Table 5 shows the position of the individual bands and the intensity ratios of the D and G peaks (I_D_/I_G_).

By analyzing the ratio of the I_DA_/I_GA_ peak intensities read from the Raman spectrum, the level of the defect in the graphene structure can be characterized. The I_D_/I_G_ intensity ratio for rGO is 0.89, which may indicate that, despite the GO reduction process, some carbon–oxygen bonds still remained in its structure.

For the samples subjected to functionalization, the I_D_/I_G_ ratio increases with the extension of the functionalization time, which may indicate a progressive disturbance of the graphene lattice and an increase in the volume of defects with sp^3^ hybridization.

### 3.4. SEM

Figure 6 presents selected SEM images of functionalized graphene materials. An irregular, porous structure with a multilayer structure, composed of overlapping wrinkled layers with a developed surface, was observed. On the outer surfaces Figure 6a), the occurrence of irregular particles with sizes from 0.5 to 10 µm was observed, which are most likely the effect of doping the graphene structure and the formation of defects with sp^3^ hybridization. Moreover, a large amount of microporosity and narrow gaps between the layered structures can be observed on the surface (Figure 6b).

### 3.5. Analysis of Specific Surface Area (BET)

The selected samples were analyzed for their textural properties: BET (specific surface area), total pore volume and average pore size (radius) by measuring low-temperature nitrogen adsorption/desorption. The calculation range of p/p° was 0.05 to 0.3, and the obtained results are presented in Table 6. The distribution of N_2_ adsorption and desorption isotherms, as well as pore size distribution are presented in Figure 7. The shape of the adsorption and desorption isotherms and the pore size distribution of all tested samples are very similar, which proves them being of similar nature. The initial section of the adsorption isotherm (p/p° = 0, Figure 7) is characteristic of microporous materials; however, its further course is typical for the type II isotherm (classification according to IUPAC) with a hysteresis loop from p/p° = 0.45. Moreover, according to the classification, the shape of the hysteresis loop corresponds to the H4 type associated with pores with the shape of narrow gaps formed between two planes and the presence of micropores, whose presence is confirmed on the pore size distribution graph.

### 3.6. Determination of Electrochemical Properties by Cyclic Voltammetry

The ability of electric charge storage was determined on the basis of cyclic voltammetry. The material tested showed a capacity between 450 and 550 mAh * g^−1^ (working up to 2.5 V). In practice, the anode materials for lithium-ion cells do not work in the range greater than 1.5 V. In this case, the capacity in this range is less than 300 mAh * g^−1^, which is less than the value assigned to commercial graphite anodes (max. 372 mAh * g^−1^).

Figure 8 presents the graphs for the first and fifth cycles of cyclic voltammetry for the tested materials, and comparatively for unmodified material (marked 12MN). As can be seen, the nature of the processes (as indicated by the shape and course of the curves) is very similar for all three samples. Comparing the tested samples with the starting material, it can be seen that the structure of the material has changed during the functionalization. The obtained curves indicate a material with a disordered structure, as opposed to graphite or base material. The synthesized materials show capacitive properties resulting from the expansion of the specific surface area.

The peaks visible in the graphs (0.005–0.3 V vs. Li/Li^+^) indicate reactions of attached functional groups or structures formed with lithium. These peaks repeat in each cycle, which indicates that the reaction is reversible.

We also observe a large loss of capacity during the first cycle of operation. This is due to the reaction of some attached functional groups (e.g., fluorine) with lithium during the insertion process (reactions taking place in primary cells) and due to the formation of a solid electrolyte interphase (so called SEI) layer (peaks at c.a. 0.55 V vs. Li/Li^+^). The decrease in the capacity associated with the SEI formation is mainly caused by the expansion of the specific surface area of the material after the functionalization process [48,49].

## 4. Conclusions

This paper presents the hydrothermal process of doping reduced graphene oxide with boron and fluorine atoms with the use of BF_3_·THF, an organic chemical compound from the group of cyclic ethers.

The conducted analyses confirmed that the graphene structure was successfully doped with fluorine atoms, as well as an addition in the form of boron atoms.

The highest intensity of the C-F bond vibrations peaks was observed for the samples modified with BF_3_·THF with a concentration of 1.5% for the reaction times of 24 and 36 h, and with a higher modifier concentration of 3%, the stronger effect was obtained after 16 h.

Functionalization led to an increase in the value of the I_D_/I_G_ ratio, along with the extension of the functionalization time, which may indicate a progressive disturbance of the graphene lattice and an increase in the volume of defects with sp^3^ hybridization.

The research confirmed that the structure of the tested material changed during functionalization. The obtained results indicated a material with a disordered, multilayered structure, consisting of overlapping, wrinkled layers with a developed surface.

The synthesized materials showed a theoretical specific capacity between 450 and 550 mAh * g^−1^ (0.005–2.5 V). The decrease in capacity during the first cycle of operation is a result of the reaction of some of the attached functional groups with lithium during the insertion process and the expansion of the specific surface area of the material.

Due to the fact that the FTIR and Raman analyses showed that, despite the GO reduction process, some carbon–oxygen bonds still remained in its structure, it is necessary to conduct further research in order to eliminate the resulting limitations, taking into account the change of the material reduction mechanism.

## Figures and Tables

**Figure 1 materials-14-00679-f001:**
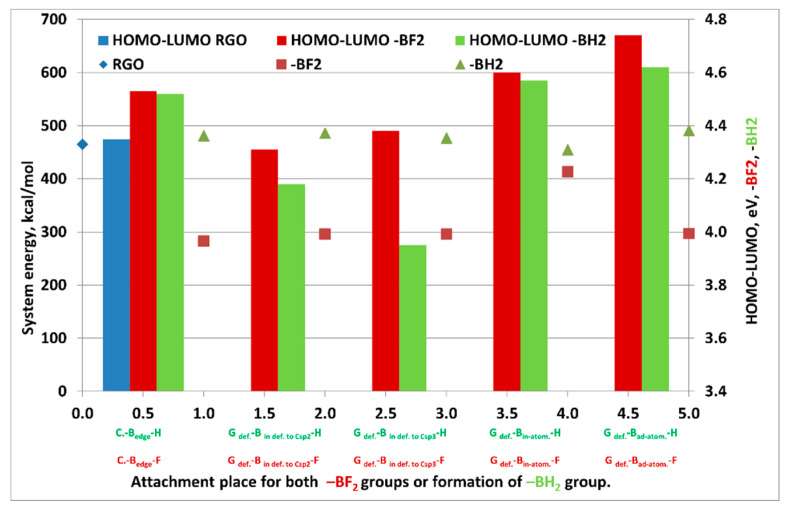
Analysis of the energy value of graphene systems depending on the place of attachment of -BF_2_ groups or the formation of -BH_2_ groups, respectively, from the left: on the edge of the flake (1), in the area of the defect in the form of vacancy of 1 C atom to carbon with sp^2^ (2) or sp^3^ (3) hybridization, either in-atomically for carbon sp^2^ hybridizing the graphene structure (4) or ad-atomically (5) with respect to the unmodified, defective graphene flake.

**Figure 2 materials-14-00679-f002:**
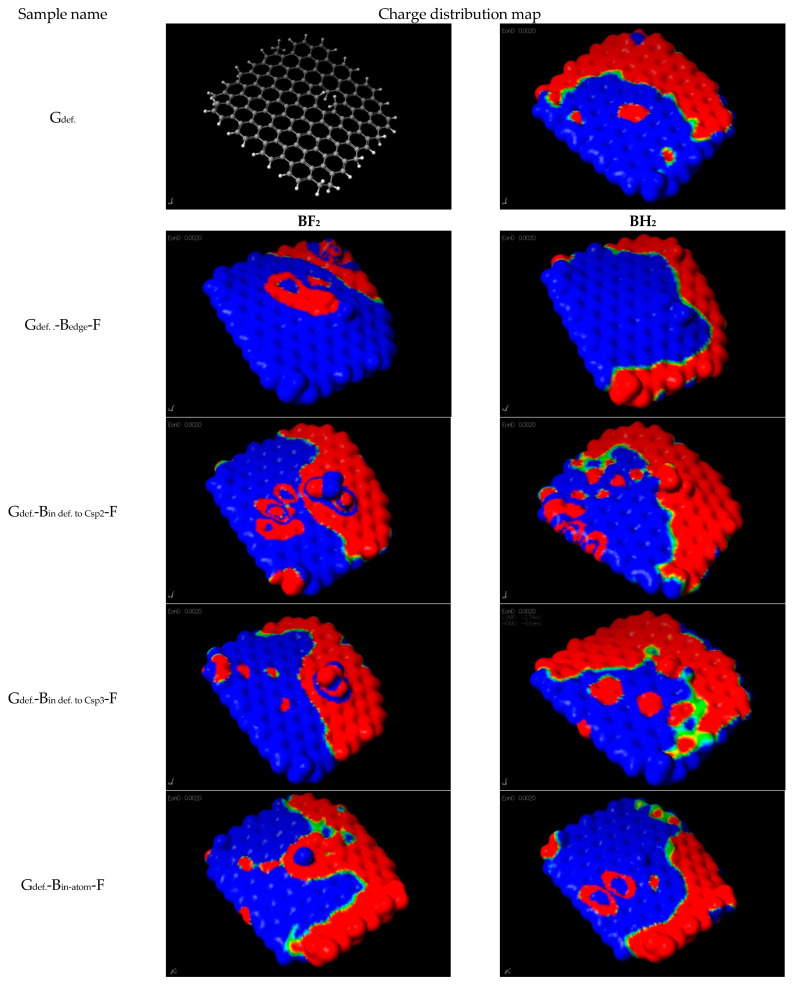
Comparison of distribution of electrostatic potential of the analyzed graphene systems depending on the place of attachment of -BF_2_ groups or the formation of -BH_2_ groups, according to the nomenclature described in Table 2.

**Figure 3 materials-14-00679-f003:**
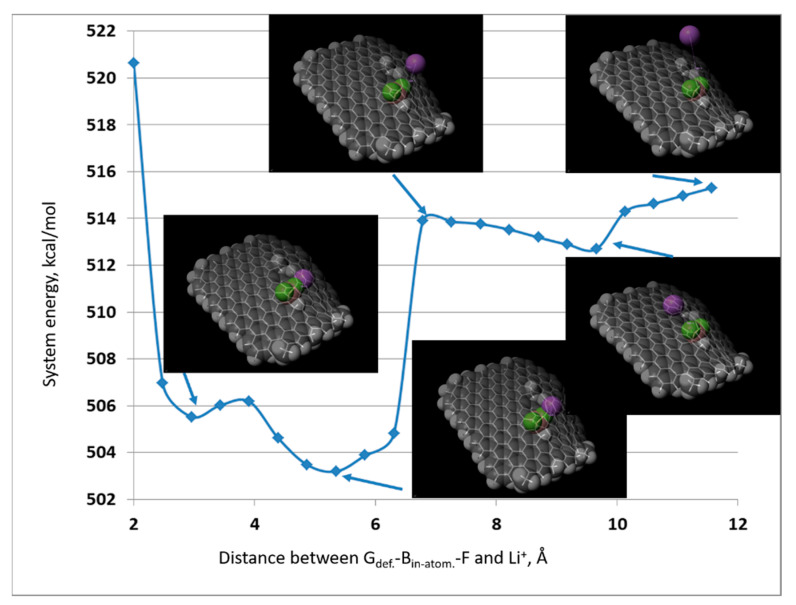
The interaction energy of lithium ions with the structure of Graphene functionalized with -BF_2_ group (G_def._-B_ad-atom_-F—according to the nomenclature described in Table 2).

**Figure 4 materials-14-00679-f004:**
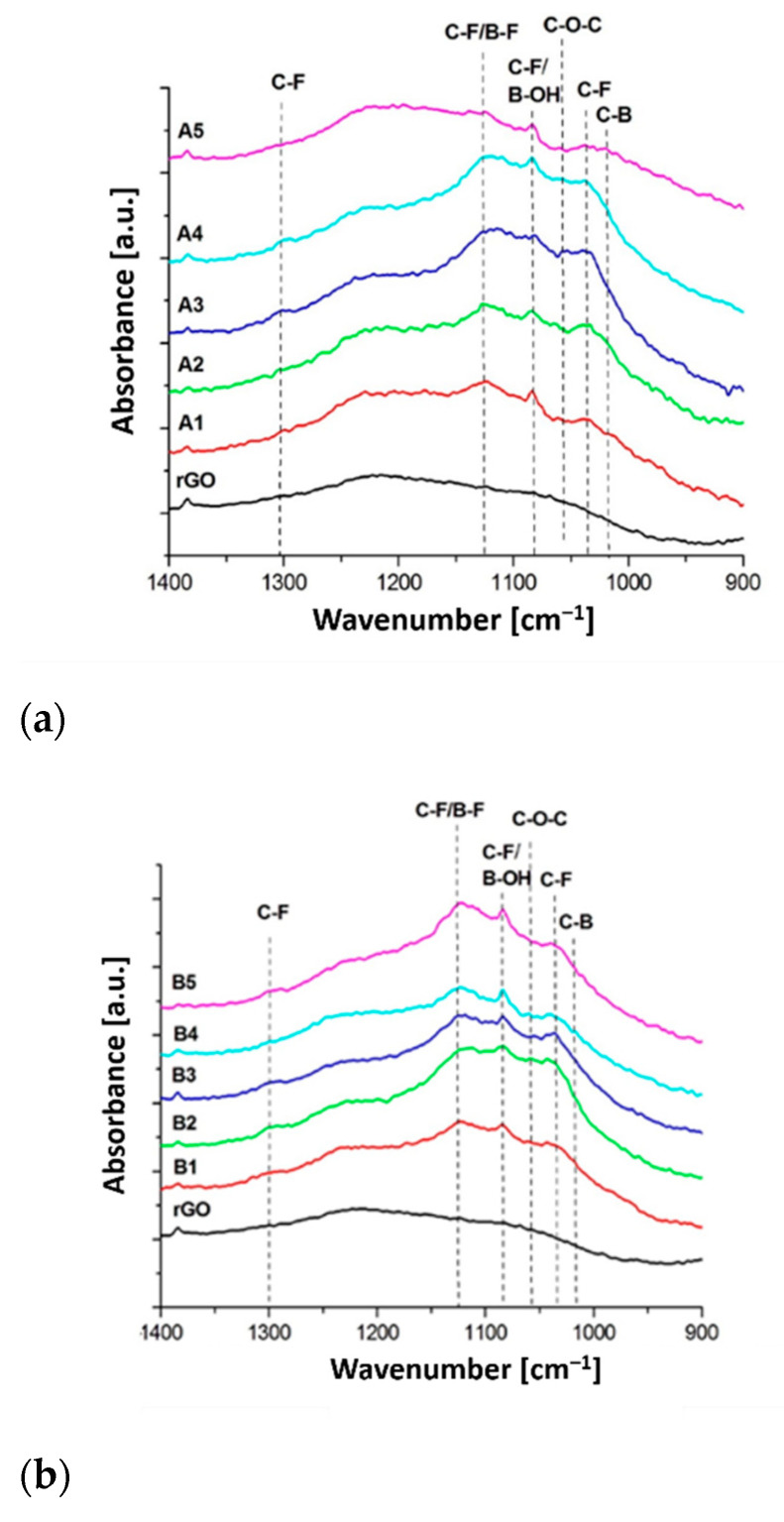
FTIR spectra of rGO without modification and functionalization with BF_3_·THF at 1.5% concentration (samples A1–A5) (**a**) and 3% concentration (samples B1–B5) (**b**) with relevant wavenumbers highlighted, and the full analyzed spectra for both cases (**c**) for samples A1–A5, (**d**) for samples B1–B5.

**Figure 5 materials-14-00679-f005:**
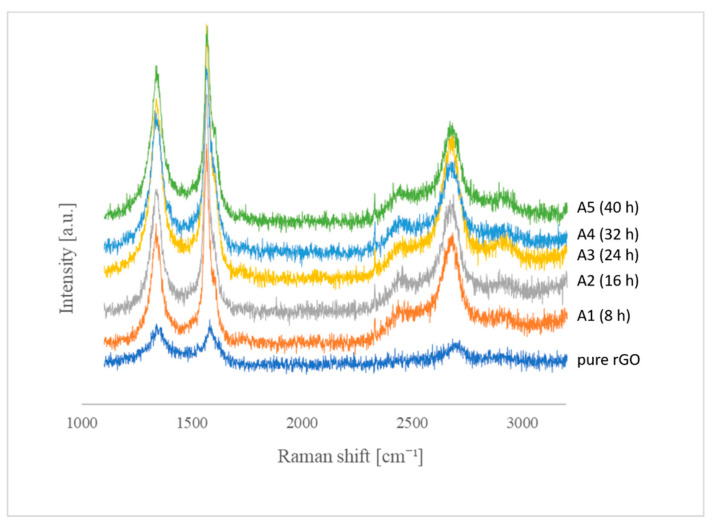
Summary of Raman spectra for the material before modification (rGO) and the obtained end products (samples A1–A5).

**Figure 6 materials-14-00679-f006:**
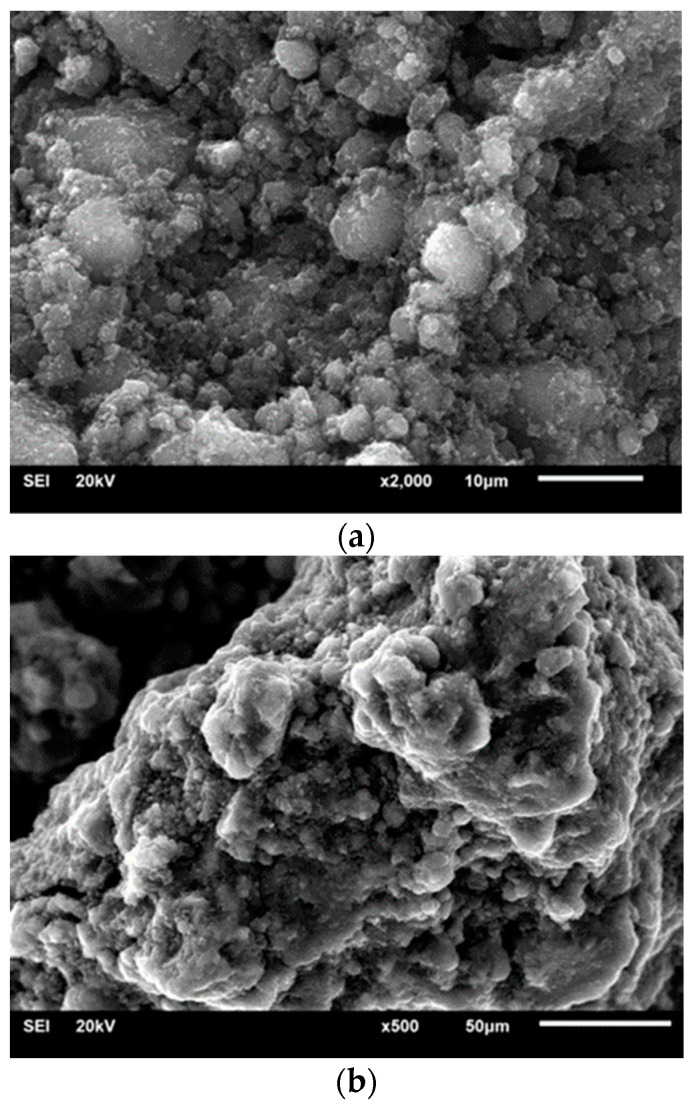
SEM images of the obtained doped graphene materials—sample A1 (**a**) and A5 (**b**).

**Figure 7 materials-14-00679-f007:**
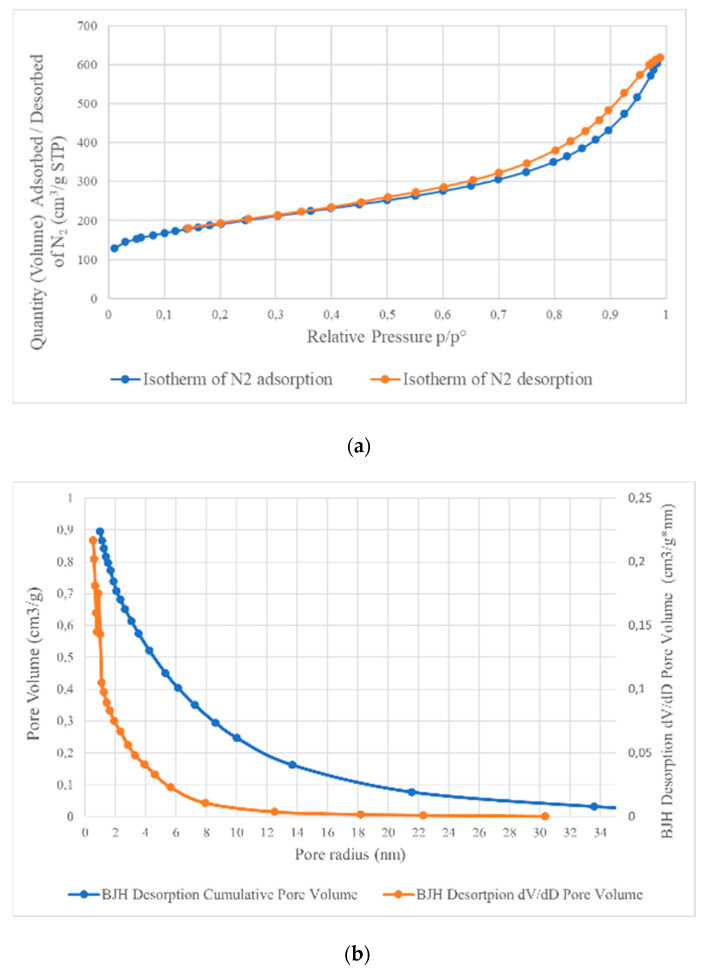
Adsorption–desorption isotherms of N_2_ for A_1_ sample (**a**) and its pore sizes (**b**).

**Figure 8 materials-14-00679-f008:**
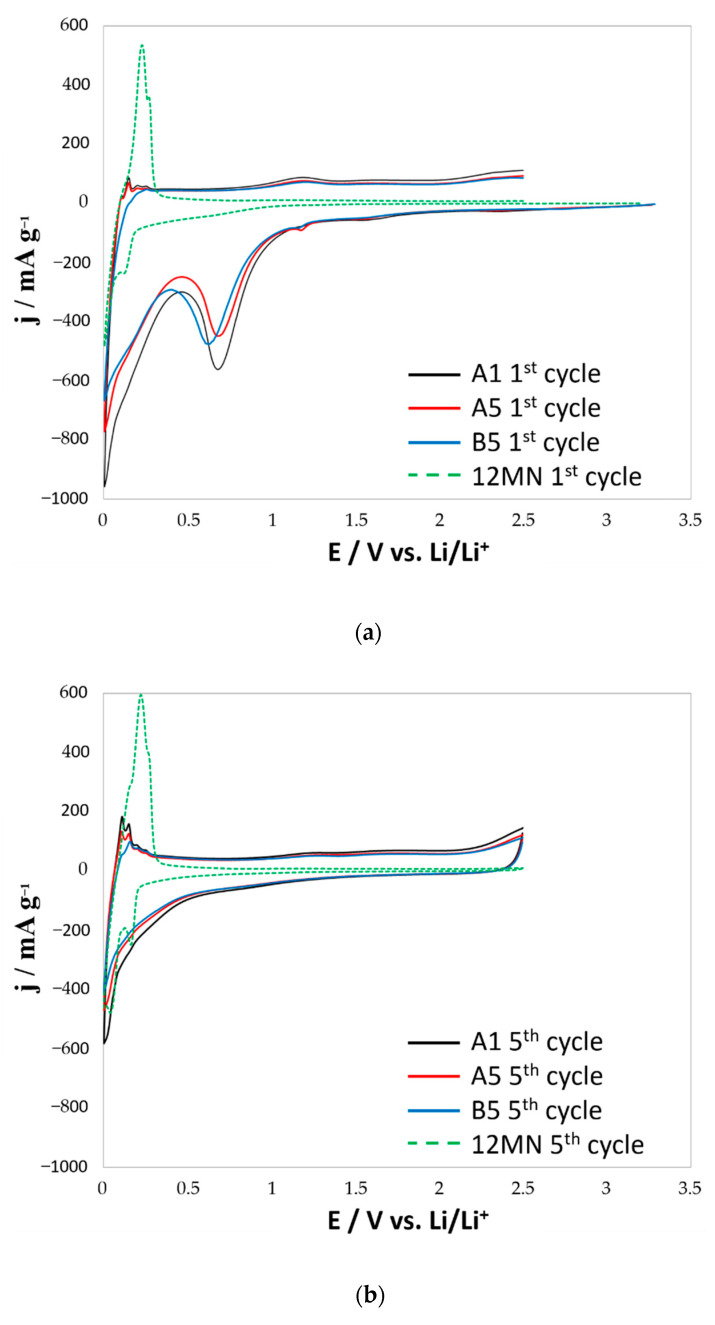
The first (**a**) and fifth (**b**) cycles of cyclic voltammetry for the tested materials and comparatively for the unmodified material (12NM).

**Table 1 materials-14-00679-t001:** Sample naming depending on the functionalization time and concentration of boron fluoride tetrahydrofuran (BF_3_·THF)_._

		Functionalization Time (h)				
		24	48	72	96	120
BF_3_·THF concentration (%)	1.5	A_1_	A_2_	A_3_	A_4_	A_5_
	3	B_1_	B_2_	B_3_	B_4_	B_5_

**Table 2 materials-14-00679-t002:** Designations of the analyzed graphene systems depending on the place of attachment of -BF_2_ groups or formation of -BH_2_ groups.

Sample Name	Sample Description
G_def._	A model graphene flake made of 149 C atoms and 46 hydrogen atoms. Additionally, the model has a structural defect in the form of a vacancy of one C atom.
G_def._-B_edge_-F	As a result of reaction with a hydrogen substitution reaction (on the edge of the flake) with BF_3_·THF, a C_graphene_-BF_2_ group is formed
G_def._-B_in def. to Csp2_-F	As a result of a hydrogen substitution reaction at carbon C_sp2_ (in the graphene flake defect) with BF_3_·THF, a C_graphene_-BF_2_ group is formed
G_def._-B_in def. to Csp3_-F	As a result of a hydrogen substitution reaction at carbon C_sp3_ (in the graphene flake defect) with BF_3_·THF, a C_graphene_-BF_2_ group is formed
G_def._-B_in-atom_-F	As a result of a reaction of C_sp2_ substitution in the graphene with BF_3_·THF, a -C_graphene_-B-F group is formed.
G_def._-B_ad-atom_-F	As a result of a reaction of C_sp2_ substitution in the graphene with BF_3_·THF, a C_graphene_-BF_2_ group is formed.
G_def._-B_edge_-H	As a result of a hydrogen substitution reaction (at the edge of a flake) with BF_3_·THF, a C_graphene_-BH_2_ is formed.
G_def._-B_in def. to Csp2_-H	As a result of a hydrogen substitution reaction at carbon C_sp2_ (in the graphene flake defect) with BF_3_·THF, a C_graphene_-BH_2_ group is formed.
G_def._-B_in def. to Csp3_-H	As a result of a hydrogen substitution reaction at carbon C_sp3_ (in the graphene flake defect) with BF_3_·THF, a C_graphene_-BH_2_ group is formed.
G_def._-B_in-atom_-H	As a result of a reaction of C_sp2_ substitution in the graphene with BF_3_·THF, a -C_graphene_-B-H group is formed.
G_def._-B_ad-atom_-H	As a result of a reaction of C_sp2_ substitution in the graphene with BF_3_·THF, a -C_graphene_-BH_2_ group is formed.

**Table 3 materials-14-00679-t003:** Highest (HOMO) and Lowest (LUMO) Occupied Molecular Orbital analysis and the energy of graphene systems depending on the place of attachment of -BF_2_ groups or formation of -BH_2_ groups, respectively: on the edge of the flake (a), in the area of the defect in the form of vacancy 1 C to carbon with sp^2^ (b) or sp^3^ hybridization (c), either atomically for the sp^3^ hybridized carbon of the graphene structure (d) or adatomically (e) with respect to an unmodified, defective graphene flake.

Sample Name	HOMO, eV	LUMO, eV	ΔE = E_HOMO_ − E_LUMO_, eV	System Energy, kcal/mol
**G_def._**	−6.77	−2.42	4.35	465.0
**G_def._-B_edge_-F**	−6.83	−2.30	4.53	283.0
**G_def._-B_in def. to Csp2_-F**	−6.72	−2.41	4.31	296.0
**G_def._-B_in def. to Csp3_-F**	−6.80	−2.42	4.38	296.0
**G_def._-B_in-atom_-F**	−7.00	−2.40	4.6	413.0
**G_def._-B_ad-atom_-F**	−6.97	−2.23	4.74	297.0
**G_def._-B_edge_-H**	−6.82	−2.30	4.52	481.0
**G_def._-B_in def. to Csp2_-H**	−6.60	−2.42	4.18	486.0
**G_def._-B_in def. to Csp3_-H**	−6.69	−2.74	3.95	477.0
**G_def._-B_in-atom_-H**	−6.93	−2.36	4.57	455.0
**G_def._-B_ad-atom_-H**	−6.85	−2.23	4.62	491.0

**Table 4 materials-14-00679-t004:** The areas of the peaks derived from groups containing in their structure atoms of fluorine and boron.

Sample Name	Wavenumber (cm^−1^)					
	1310–1280 cm^−1^ (C-F (*s*) in CF-CH_3_ and C-F (*as*) in CF_2_	1126 cm^−1^ C-F (*ss*) in CF_2_ /B-F (*s*)	1085 cm^−1^ (C-F (*s*) in CF)/B-OH (*s*)	1055 cm^−1^ (C-O-C *as*))	1035 cm^−1^ (C-F (*s*) from CF)	1018 cm^−1^ (C-B)
**A_1_**	-	0.21	0.06	-	0.14	0.04
**A_2_**	-	0.25	0.06	0.01	0.22	0.06
**A_3_**	0.02	0.27	0.03	0.02	0.20	0.04
**A_4_**	0.02	0.22	0.05	-	0.19	0.05
**A_5_**	0.01	0.12	0.05	0.01	0.14	0.03
**B_1_**	0.02	0.09	0.05	0.01	0.10	0.02
**B_2_**	0.04	0.08	0.05	0.01	0.14	0.03
**B_3_**	0.03	0.15	0.07	-	0.20	0.05
**B_4_**	0.01	0.20	0.09	-	0.19	0.08
**B_5_**	0.02	0.26	0.10	-	0.21	0.12

**Table 5 materials-14-00679-t005:** Summary of the positions of individual bands and the values of the I_DA_/I_GA_ and intensity ratios of the D and G (I_D/_I_G_) ratios.

Parameter	0	A_1_	A_2_	A_3_	A_4_	A_5_	Band
Area	41,684.09	58,487.56	64,284.79	11,1025.87	95,620.83	101,770.27	Peak D
FWHD	124.56	54.29	53.60	65.57	73.29	81.31	
Intensity(A.U)	326.76	872.73	1008.73	1354.54	1048.21	1261.63	
Raman shift	1341.58	1339.32	1339.28	1340.39	1340.43	1341.41	
Area	43,444.72	40,909.61	39,571.78	64,445.61	37,184.21	39,406.77	Peak G
FWHD	143.15	23.04	20.52	30.89	24.81	27.14	
Intensity(A.U)	363.95	1387.60	1471.78	1603.80	1185.59	1439.51	
Raman shift	1580.70	1566.53	1566.06	1568.22	1567.30	1570.73	
Area	33,384.10	122,004.23	130,523.80	148,816.48	101,785.16	128,345.60	Peak 2D
FWHD	203.60	127.93	135.84	114.73	123.89	146.59	
Intensity(A.U)	226.35	779.66	856.84	1101.80	744.65	879.08	
Raman shift	2698.19	2674.78	2674.91	2675.59	2675.32	2682.03	
Area	-	38,543.12	41,647.04	33,785.90	49,792.27	58,367.66	Peak D’
FWHD	-	94.98	72.45	69.10	67.66	71.98	
Intensity(A.U)	-	426.52	610.73	586.99	689.81	822.08	
Raman shift	-	1604.59	1594.56	1599.18	1598.95	1602.38	
Area	-	28,219.95	17,986.16	18,834.84	25,609.52	32,190.22	Peak D+G
FWHD	-	118.16	76.25	88.23	107.32	78.18	
Intensity(A.U)	-	342.23	395.32	411.94	373.54	412.70	
Raman shift	-	2460.62	2451.43	2458.03	2455.01	2921.88	
ID_A_/IG_A_	0.96	1.43	1.62	1.72	2.57	2.58	
ID_I_/IG_I_	0.89	0.63	0.69	0.84	0.88	0.88	

**Table 6 materials-14-00679-t006:** Textural properties of the samples.

Sample	Specific Surface Area (BET), m^2^/g	Total Pore Volume (BJH), cm^3^/g	Average Pore Size (Radius) (BJH), nm
**A_1_**	654	0.895	3.49
**B_1_**	638	0.909	3.61
**A_5_**	660	0.909	3.44
**B_5_**	586	0.885	3.60

## Data Availability

The data presented in this study are available on request from the corresponding author.

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
