# Peer review of "Functionalization Mechanism of Reduced Graphene Oxide Flakes with BF3·THF and Its Influence on Interaction with Li+ Ions in Lithium-Ion Batteries"

_materials, 2021, doi:10.3390/ma14030679_

Round 1
Reviewer 1 Report
Reviewers' comments:
Manuscript ID: materials-1051184
Full Title: Functionalization mechanism of reduced graphene oxide flakes with THF-BF3 and its influence on interaction with Li+ ions in lithium-ions batteries.
Comments:
The manuscript reported on Functionalization mechanism of reduced graphene oxide flakes with THF-BF3 and its influence on interaction with Li+ ions in lithium-ions batteries. The manuscript needs a detailed editing. The authors need to provide answers to the issues listed below before the manuscript should be accepted for publication.
- Some sentences need reconstruction and the level of English should be improved.
- In the Abstract: the authors need to improve with more specific short results.
- Line number 25 - m2/g to m2/g
- Keywords: the authors need to improve with more specific short keywords.
- Line number 70 - Check BF3 or BF3
- Line number 75 - temperature (90˚C, 120˚C, 150˚C, 180˚C) for 24h…… change; temperature (90˚C, 120˚C, 150˚C and 180˚C) for 24h.
- Line number 84 - H2SO4-H3PO4 to H2SO4-H3PO4
- Line number 75 - 24 h, and - Line number 84 - 2 hours; make clear (h or hours)
- Figure 4, not clear make clear.
- Figure 5, make clear understanding
- Figure 8, make clear understanding
- Conclusion should be concise.
- Several faults: are added or missing spaces between words: see manuscript file.
- References: there are recent references in 2019 and 2020 treating the same subject, you can use and make all references in same format for volume number, page numbers and journal name, because it is difficult to searching and reading.
So that I recommended this manuscript to major revision and for future process.
Author Response
Please see the attachement

Reviewer 2 Report
Comments on “Functionalization mechanism of reduced graphene oxide flakes with THF-BF3 and its influence on interaction with Li+ ions in lithium-ions batteries”
1- There are minor typos in the paper that authors should address and fix in the next version.
2- Are there any advantages when comparing the method used by the authors with the literature? This should be clarified.
3- Apart from batteries, is there any other applications for the functionalised graphene?
4- Introduction section: authors could review some of the potential applications of graphene and carbon-based nano-suspensions in thermal engineering and advanced transport phenomena to reflect the importance of their work. For example, searching the literature, following papers are suggested to be read and used to enrich the introduction part:
- Thermal assessment of nano-particulate graphene-water/ethylene glycol (WEG 60: 40) nano-suspension in a compact heat exchanger
- Thermal evaluation of graphene nanoplatelets nanofluid in a fast-responding HP with the potential use in solar systems in smart cities
- Diurnal thermal evaluation of an evacuated tube solar collector (ETSC) charged with graphene nanoplatelets-methanol nano-suspension
5- Fig. 8: there is an inflection point (a minima) seen at x=0.55. Can authors justify the physics behind these phenomena. Also, it would be more interesting if authors report A.A.D.% for A1, A5, B5 for all cases shown in Fig. 8.
6- A paragraph should be added to the introduction section showing the gap, objective and also the potential novelty of this work in a clear way. It is still ambiguous in the paper.
All in all, this is a very interesting paper and it can be published once above comments are addressed.
Author Response
Please see the attachement

Reviewer 3 Report
The paper by Kaczmarek et al. presented the hydrothermal process of doping reduced graphene oxide with the use of THF-BF3. Several characterization methods were used to confirm the structural change after doping and functionalization. The results indicated a materials with a disordered, multi-layered structure which showed high specific capacity of 450-550 mAhg-1. Basically, the whole manuscript is well organized and most of the experimental results are reasonable. I would recommend publishing after minor revision. The followings are necessary to be addressed before the publication process is completed. 1) Author used short paragraphs throughout the paper (especially introduction and conclusion section). Some paragraphs only have one sentence (ie. Page 14, Line 371). Please consider reorganizing them into larger paragraphs. 2) The substrate for this study is not clearly recognized. What kind of substrate is used for doping study? 3) Please comment on the chemical stability of doped graphene. Can this dopant sustain well to acid or base chemicals? 4) How is the carrier concentration (n) of doped graphene compared to that of pristine graphene? I suggest author to carry out Hall-effect measurements and FET to extract carrier concentration. 5) Page 2, Line 49. "lithium-ion (LiB) batteries" should written as "lithium-ion batteries (LiB)" 6) Page 5, Figure 1. Subcaption (a,b,c,d,e) from description was not found in the figure. Please clarify. 7) Page 8, Line 200-207. Author mentioned FTIR peaks at 3440cm-1, 1450 cm-1, 1720 cm-1 and 1580 cm-1 with reference to Figure 4. However, the Figure 4 spectra only showed the wavelength range from 900 cm-1 to 1400 cm-1. Please add the Figure with full x-axis scale from 900cm-1 to 4000cm-1.Author Response
Please see the attachement

Round 2
Reviewer 1 Report
The manuscript can published. The authors have answered the questions.
Author Response
Thank you very much for accepting the revised version of the manuscript.
Reviewer 2 Report
The author did not answer my questions accordingly. They just have tried to convince me that my comments are wrong and not applicable! I give one more chance to authors to answer all of my questions. If they cannot answer all of my questions, I have no choice just to reject the paper.
Author Response
Please see the attachement
